# Silkworm Cocoon: Dual Functions as a Traditional Chinese Medicine and the Raw Material of Promising Biocompatible Carriers

**DOI:** 10.3390/ph17070817

**Published:** 2024-06-21

**Authors:** Zhijie Tian, Chuncao Zhao, Ting Huang, Lining Yu, Yijie Sun, Yian Tao, Yunfeng Cao, Ruofei Du, Wenhui Lin, Jia Zeng

**Affiliations:** 1School of Chemistry & Environmental Engineering, Shanghai Institute of Technology, Shanghai 201418, China; 226061216@mail.sit.edu.cn; 2NHC Key Laboratory of Reproduction Regulation, Shanghai Engineering Research Center of Reproductive Health Drug and Devices, Shanghai Institute for Biomedical and Pharmaceutical Technologies (SIBPT), Shanghai 200032, China; zhaochuncao@sibpt.com (C.Z.); huangting@sibpt.com (T.H.); yulining@sippr.org.cn (L.Y.); taoyian@sibpt.com (Y.T.); caoyunfeng@sibpt.com (Y.C.); 3Huadong Hospital Affiliated to Fudan University, Shanghai 200040, China; sunyijie@fudan.edu.cn; 4Innovation Research Institute of Traditional Chinese Medicine, Shanghai University of Traditional Chinese Medicine, Shanghai 201203, China; drf790101@shutcm.edu.cn

**Keywords:** silkworm cocoon, silk fibroin, silk sericin, biological activity, biocompatible carriers, applications

## Abstract

The silkworm cocoon (SC), both as a traditional Chinese medicine and as the raw material for biocompatible carriers, has been extensively used in the medical and biomedical fields. This review elaborates on the multiple functions of SC, with an in-depth analysis of its chemical composition, biological activities, as well as its applications in modern medicine. The primary chemical components of SC include silk fibroin (SF), silk sericin (SS), and other flavonoid-like bioactive compounds demonstrating various biological effects. These include hypoglycemic, cardioprotective, hypolipidemic, anti-inflammatory, antioxidant, and antimicrobial actions, which highlight its potential therapeutic benefits. Furthermore, the review explores the applications of silk-derived materials in drug delivery systems, tissue engineering, regenerative medicine, and in vitro diagnostics. It also highlights the progression of SC from laboratory research to clinical trials, emphasizing the safety and efficacy of SC-based materials across multiple medical domains. Moreover, we discuss the market products developed from silk proteins, illustrating the transition from traditional uses to contemporary medical applications. This review provides support in understanding the current research status of SC and the further development and application of its derived products.

## 1. Introduction

The silkworm cocoon (SC) is the dry shell of *Bombyx mori* L., which belongs to the Saturniidae family. SC is widely used in the silk industry, agriculture, biological materials, medicine, and other fields. In medicinal applications, SC was first mentioned in the famous medical book *Compendium of Materia Medica* and has been used as a traditional medicine for hundreds of years in China for its diverse therapeutic effects [1,2,3]. SC has pharmacological effects to collect astringency, stop bleeding, quench thirst, and detoxify boils. Hence, it is used for the treatment of hematemesis, bloody stools, metrorrhagia, excessive urination, and the pathogenesis of carbuncle and pus.

Modern research indicates SC contains major chemical components such as silk sericin (SS) and silk fibroin (SF) [4,5]. These components endow SC with significant biological activities, not only enhancing the traditional health benefits, but also providing foundational properties for innovative therapeutic techniques. The therapeutic actions of SS and SF are extensive, including antidiabetic effects [6,7], cardioprotective properties [8,9,10], and lipid-lowering capabilities [11]. Concurrently, the ability to modulate inflammatory responses [12,13,14] and resist oxidative stress [15,16,17,18] broadens their application in disease prevention and treatment. Furthermore, the inherent antimicrobial [4,19,20] and antiviral [21] properties of these proteins enhance their effectiveness in developing cutting-edge medical interventions.

With advancements in biotechnology, the application of SC has expanded into the modern biomedical field, and SC has been proven to be a promising biocompatible carrier [22,23]. Therefore, it is considered not only to be a key component of traditional medicine but also a crucial material in contemporary medical innovation. Presently, its applications primarily manifest in areas such as drug delivery systems [24,25,26], tissue engineering [27,28,29,30], and regenerative medicine [30,31]. These studies leverage the biodegradability and excellent biocompatibility of SC to develop new medical materials and treatment methods, showcasing its potential as a natural resource in the contemporary medical domain.

Considering these developments, in this study, we deeply examine the dual functions of SC in both laboratory research and clinical studies by providing an in-depth analysis of the chemical composition and biological activity of SC and highlighting its emerging applications in biomedicine. The aim of this review is to explore how SC can be cleverly redesigned for future medical science.

## 2. Chemical Composition

The chemical composition of SC primarily consists of SF and SS. SF forms the structural framework of SC, and SS acts as an adhesive that bonds the fibrous components within the SC together. In addition to these primary components, SC contains bioactive substances such as flavonoids, peptides, and calcium oxalate crystals, as shown in Table 1.

### 2.1. Silk Fibroin

SF is the principal protein, and approximately 70% the amount of SC is characterized by a hierarchical structure of a light chain (approximately 26 kDa) and a heavy chain (approximately 350 kDa) bonded through disulfide links. The primary structure of the heavy chain consists of 12 large repeating hydrophobic domains of amino acids, such as glycine, alanine, serine, valine, and tyrosine, separated by 11 short hydrophilic regions, while the light chain features hydrophilic amino acids, such as glutamic acid, lysine, and aspartic acid [5,32]. Approximately 5.3% of SF is composed of tyrosine, while 0.2% of SF is lysine residues. Both have functional groups that can react with epoxides. The ability of SF to form β-sheet structures via physical cross-linking allows the silk materials to exhibit a range of multifunctional properties that can be finely adjusted [33]. Owing to its low immunogenicity and fibrous structure resembling collagen I, SF is one of the most attractive natural biomaterials for biomedical applications [34].

### 2.2. Silk Sericin

SS, which accounts for approximately 30% of the SC, is a glycoprotein adhesive containing 18 different amino acids primarily organized in the form of polar side chains, such as hydroxyl, acidic, and basic amino acids, SS forms three primary peptide complexes with molecular weights of 150, 180–250, and 400 kDa. These functional groups enable SS to easily cross-link, blend, and copolymerize with other polymers, thereby producing improved biocompatible materials with enhanced properties [35,36].

### 2.3. Flavonoids

Flavonoids exist in the cocoons of different silkworm species. Seven flavonoid compounds were reportedly isolated, purified, and identified from yellow–green SC, and their aglycones were isolated and purified from the ethanol extract of SC [37]. The presence of flavonoids in the SC helps to provide antioxidant protection to the SC itself [38].

### 2.4. Other Components

Many types of outdoor-reared SC contain abundant calcium oxalate monohydrate crystals, primarily comprising oxalic acid and calcium ions, with a chemical formula of CaC_2_O_4_·H_2_O [39]. The formation of these crystals may be related to the metabolic substances and environmental conditions within the silkworm, making them a common component of the SC [40].

In addition to primary proteins like SF and SS, SC is composed of peptides with different sequences and functions. These peptides are small molecular protein fragments composed of amino acids and are mainly found in the degradation products of SF or SS, which could play roles in regulating biological processes within the silk gland, contributing to the microenvironment of the SF and ensuring the silkworm’s development and transformation into a moth [41].

## 3. Biological Activities

SC exhibits a wide range of pharmacological effects, such as lowering blood glucose and lipid levels, heart protection, anti-inflammatory, antioxidative, and anti-infection properties. Moreover, it shows potential in cancer therapy. Its unique chemical composition and elastic properties facilitate wound healing and provide pain relief, making it highly applicable in the medical field. The biological activities and mechanisms of SC are summarized in Table 2.

### 3.1. Hypoglycemic Action

In traditional medicine, SC is known as an antidiabetic agent that was widely used in ancient formulas. Modern medicine has also confirmed that the primary components of SC are ideal candidates for the prevention and treatment of diabetes. SF, SS, peptides, or their hydrolysates present α-glucosidase inhibitory activity. Additionally, other hypoglycemic mechanisms of these compounds include the inhibition of the expression of intestinal glucose transporters, the promotion of the regeneration of pancreatic β-cells, and the activation of the insulin-like signaling pathway [42]. The ethanolic extract from the green cocoon layer of the silkworm has been demonstrated to improve glucose metabolism by reducing inflammatory reactions, enhancing the antioxidant capacity and insulin sensitivity, and regulating the balance between glycolysis and gluconeogenesis [43].

Diabetes can also cause many disorders such as nephropathy, neuropathy, and retinopathy. SC has significant therapeutic effects on a variety of complications caused by diabetes. The ethanolic extract is rich in quercetin and kaempferol, and the glucosides from SC reduced blood glucose levels and improved body weights in diabetic mice, and it may have a potential therapeutic application in the treatment of diabetic nephropathy. The ethanolic extract inhibits the expressions of renal tumor necrosis factor-alpha, monocyte chemoattractant protein-1, fibronectin, and P38 mitogen-activated protein kinase. Furthermore, it significantly elevates the levels of superoxide dismutase and glutathione peroxidase in diabetic mice [39].

There is also evidence that diabetes affects men’s sexual function through changes in the process of spermatogenesis or the endocrine changes affecting it; it affects erectile dysfunction and decreases libido, it can lessen sperm DNA damage, and it affects the hypothalamic–pituitary–gonadal axis [44]. A recent study suggested that a hydroalcoholic extract of SC decreased the effects of diabetes on the hypothalamic–pituitary–gonadal axis. It increased the level of sex hormone secretion in a dose-dependent manner in diabetic rats by affecting the activity of the pituitary–gonadal axis. Additionally, the dose-dependent administration of the aqueous extract of SC can improve testicular tissue injuries in type I diabetic rats, thus protecting fertility [45]. These findings provide evidence of SC as a potential therapeutic approach for the treatment of diabetes and insulin resistance.

### 3.2. Cardioprotective Effect

In terms of cardiac protection, SS plays a critical role in myocardial injury, as the protective properties of amino acids have just been recognized as the fundamental reason for enhancing cardiac protection [46]. In an isoproterenol-induced rat model, SS showed a decrease in edema, the dilation of capillaries, scar formation, and yellowing of the heart.

The intraventricular septum thickness, right ventricular wall thickness, and left ventricular wall thickness decreased dramatically after SS treatment. SS also significantly increased the non-enzymatic antioxidant markers in the serum and heart tissue, including glutathione, vitamin E, and vitamin C. The results were the same for the enzymatic antioxidant marker, mitochondrial enzymes, and protein. SS reduced the synthesis of collagen in myocardiocytes, thus reducing the incidence of fibrosis, and the synthesis of collagens. In summary, it showed potent cardio-protective properties by decreasing inflammatory reactions and oxidative stress, thereby leading to improved myocardial activity and reduced cardiac damage after myocardial ischemia [9].

Mitochondrial abnormalities in myocardial cells are one of the causes of heart failure [47]. SS might improve the dysmorphic mitochondrial structure, metabolism, and energy production of cardiac mitochondria under hypercholesterolaemic conditions. The mechanisms included the upregulation of optic atrophy 1 and the reduction of NADH–ubiquinone oxidoreductase 75-kDa subunit expression, as well as an improvement in mitochondrial energy production by upregulating acetyl-CoA acetyltransferase and the NADH dehydrogenase 1a subcomplex subunit 10 expression [48].

Furthermore, the cardioprotective activity of the ethanolic extract of SC has also been revealed. It significantly prevented isoproterenol-induced myocardial damage and hypertrophy and decreased the levels of the myocardial enzyme markers [8]. The cardioprotective effect of a formulation combined with the methanol extract of SC, flaxseed oil, and coenzyme Q10 against doxorubicin induced myocardial toxicity, and this was evaluated in rats. The test formulation lowered the increase in heart weight due to hypertrophy while significantly reducing the serum levels of aspartate aminotransferase, alanine aminotransferase, lactate dehydrogenase, creatinine, triglyceride, low-density lipoprotein cholesterol, and very-low-density lipoprotein cholesterol and triglyceride. It also increased the levels of high-density lipoprotein cholesterol and antioxidant parameters, such as superoxide dismutase and tissue glutathione as a catalase. These results strongly indicated the cardioprotective effect of the SC-based formulations, which might be further developed as good alternatives for the treatment of heart-related diseases [49].

### 3.3. Hypolipidemic Activity

It has been documented in Unani medicine that SC has a protective role in hyperlipidemia that is related to its antioxidant compounds [50]. After four weeks of treatment, the crude extract of SC successfully lowered the rise of serum lipid levels caused by cholesterol-rich foods in New Zealand white rabbits and reduced the size of atherosclerotic plaques. The researchers concluded that an extract of SC could probably inhibit lipid implantation in the injured arterial wall due to its lipid lowering and antioxidant property [51].

In addition, previous studies showed that combining the SS extracted from SC and a high-fat diet intake has been associated with hypolipidemic effects [52]; therefore, SS is further considered a potential obesity prevention approach. Obesity, especially cases caused by high-fat and/or high-calorie diets, has been shown to affect intestinal wall components [53]. A high-fat diet was induced in a C57BL/6 obese mice model, and although SS treatment was not able to reverse plasma and biometric changes promoted by obesity, it restored the jejunal morphometry, including a reduction in the intestinal wall thickness and villus height and an increase in the crypt depth. Furthermore, SS increased lipid excretion in the feces of obese mice, signifying its prospective antiobesity effects [11].

### 3.4. Anti-Inflammatory Effect

Carbonized SC has been widely used for many inflammation-related diseases. SC-based carbon dots (CDs) that were prepared using a high-temperature pyrolysis method exhibited significant anti-inflammatory bioactivity that was comparable in efficacy to established treatments like dexamethasone. CDs are a class of carbon-based nanomaterials known for their luminescent properties and zero-dimensional structure, composed solely of carbon atoms and enriched with various organic functional groups on their surface [54]. The underlying mechanism of the anti-inflammatory action of SC-CDs appears to be their capacity to inhibit interleukin-6 and tumor necrosis factor-α [3], which are central to the initiation and maintenance of inflammatory responses in the body. This modulation by SC-CDs highlights a promising therapeutic pathway for managing inflammatory conditions [55].

SS has a positive effect on the treatment of chronic inflammatory diseases [56]. It can effectively reduce the thickness of the epidermis in psoriatic skin conditions, akin to the action of known anti-inflammatory drugs such as betamethasone and calcitriol. This is primarily due to its ability to regulate key inflammatory pathways, including the downregulation of the C–C motif chemokine 20 and the JAK-signal transducers and activators of the transcription signaling pathway, thus reducing the production of inflammatory cytokines and alleviating the overall inflammatory response. In conclusion, the multiple actions of SS underscore its potential application in treating psoriatic and other skin diseases, particularly in regulating epidermal cell proliferation and the dynamics of immune responses [12].

### 3.5. Antioxidant Effect

The ethanolic extract from the green cocoons, rich in flavonoids, has been reported to have excellent antioxidant properties. The half-maximal inhibitory concentration values for 2,2-diphenyl-1-picrylhydrazyl and 1,2′-azino-bis(3-ethylbenzthiazoline-6-sulphonicacid) tests were found to be 296.95 ± 13.24 μg/mL and 94.31 ± 9.13 μg/mL, respectively. Moreover, this extract was found to reduce the level of reactive oxygen species and oxidative stress in L02 cells induced by high glucose levels [57].

In a further study of the antioxidant capacity, three fractions were isolated from SC, i.e., crude SS, purified SS, and the methanolic extract. In the Trolox equivalent antioxidant capacity assay, the methanolic extract exhibited a higher antioxidant capacity with approximately twice the Trolox equivalents than the other two fractions. Murine retinal photoreceptor cells are the cell line that is most vulnerable to oxidants and play an essential role in retinopathies that are primarily caused by oxidative stress. When cultured with the murine retinal photoreceptor cells, compared with crude SS and the purified SS fractions, the methanolic extract improved the cell viability by nearly 10 times at a concentration of 1 mg/mL and displayed maximum cell activity at a concentration of 5 mg/mL. This result indicated that this fraction significantly protected the cultured cells exposed to hydrogen peroxide at different doses [58].

Furthermore, SC increased the total collagen content in human dermal fibroblast cells that were exposed to ultraviolet A1. This was because SC primarily comprises SS and associated secondary metabolites such as polyphenols and flavonoids, which may be able to partially prevent ultraviolet radiation-induced photoaging [59].

### 3.6. Antiviral and Antimicrobial Effects

The 95% ethanol extract of SC showed a high potential activity against the herpes simplex virus (HSV)-1 and HSV-2. HSV-2-infected HeLa cells treated with the 95% ethanol extract of SC drastically reduced cell death and prevented inflammation by reducing the production of inflammatory cytokine genes. The bioactive components of this portion of the extract were further shown to be gallic acid, flavonoids, and xanthophyll, and they played important roles in the antiviral activity. This study indicated that the SC extract has potential as a therapeutic treatment for herpes simplex virus (HSV) infection [21].

The underlying mechanism of the antimicrobial effects of SC was also investigated. SC contains a large number of proteins that inhibit the growth of fungi and bacteria, including protease inhibitors and seroins [60]. BmSPI51 is the most abundant SC protease inhibitor, and it has been reported to play a part in the antimicrobial role of SC. It strongly suppressed the sporular growth of three fungal species, i.e., *Candida albicans, Beauveria bassiana*, and *Saccharomyces cerevisiae*. According to the in vitro inhibition experiments, its defense mechanisms revealed that BmSPI51 inhibited fungal growth by binding to the cell wall polysaccharides, mannan, and β-glucan or preventing fungi from obtaining nutrients and retarding the rate of budding [61].

**Table 2 pharmaceuticals-17-00817-t002:** The biological activities and mechanisms of silkworm cocoon.

Biological Activities	Sample	Animal	Model	Dose	Results	Mechanisms	Refs.
hypoglycemic effect	fibroin, sericin	silkworm /in vivo	diets containing glucose or sucrose	5% added to the diets	exhibited postprandial antihyperglycemic activity	inhibition of the expression of intestinal glucose transporters, promotion of the regeneration of pancreatic β cells, or activation of the insulin-like signaling pathway	[42]
	ethanolic extract from the green cocoon sericin layer of silkworm	ICR mice /in vivo	fed with a high-fat diet and injected with streptozotocin	150, 250, and 350 mg/kg	ameliorated glucose metabolism and regulated the balance between glycolysis and gluconeogenesis	reduction of the levels of NF-κB, IL-6, and TNF-α; enhancement of the expression levels of IR, IRS, PI3K, p-Akt, and p-GSK3β involved in insulin signalling; activation of AMPK and GLUT4; reduction of the levels of G6pase and PEPCK; improvement of the GK level	[43]
	flavonoid-rich ethanolic extract from silkworm green cocoon	ICR mice /in vivo	induced by high-fat and streptozotocin	150, 250, and 350 mg/kg	regulated the glucose level and body weight and improved renal dysfunction	inhibition of the TNF-α-p38 MAP kinase signaling pathway	[39]
	hydroalcoholic extract of silk cocoon	Wistar rats /in vivo	induced by streptozotocin	200, 400, and 800 mg/kg	decreased prolactin and inhibin; increased leptin, IGF-2, activin A, insulin, LH, testosterone, FSH, and GnRH levels; improved gonadal weight, the diameter of tunica albuginea, and seminiferous tubules as well as increased the numbers of spermatocytes and Sertoli–Leydig cells	NA	[44]
cardioprotective effect	sericin	Wistar rats /in vivo	isoproterenol induced cardiac toxicity and hypertrophy	500 and 1000 mg/kg	significantly increased the non-enzymatic antioxidant markers in serum and heart tissue; significantly decreased the myocyte size	prevented the myocardial tissue from enzymatic leakage from the cell sites; reduced the synthesis of collagen in myocardiocytes, thus reducing the incidence of fibrosis; reduction of fibrosis and synthesis of collagens contributed to the protective effect against hypertrophy; decreased inflammatory reactions and oxidative stress, which led to improved myocardial activity, and reduced cardiac damage after myocardial ischemia	[9]
	sericin	Wistar rats /in vivo	cholesterol diet-induced hypercholesterolaemia model	1000 mg/kg	improved cardiac muscle contraction under hypercholesterolaemia, restored the cardiac mitochondrial structure, increased mitochondrial fusion in the heart, and inhibited the progression of apoptosis at the last stage of dysmorphic mitochondria	upregulation of OPA1 and reduction of NADH-ubiquin-one oxidoreductase 75 kDa subunit expression; improvement of mitochondrial energy production by upregulating acetyl-CoA acetyltransferase and NADH dehydrogenase 1a subcomplex subunit 10 expression	[48]
	ethanolic extract of silk cocoons	Albino Wistar rats /in vivo	isoprenaline-induced myocardial infarction	250 and 500 mg/kg	significantly prevented myocardial damage and hypertrophy, and decreased the levels of various cardiac enzymes	NA	[8]
	an emulsion formulation composed of methanol extract of silk cocoons, flaxseed oil, and coenzyme Q10	Sprague Dawley rats /in vivo	doxorubicin induced myocardial toxicity	500 mg/kg methanol extract of silk cocoons, 1.8 mL/kg flaxseed oil, and 5 mg/kg coenzyme Q10	significantly prevented the increase in serum levels of AST, ALT, LDH, and creatinine and the lipid profile, increased the levels of HDL, SOD, GSH, and CAT in heart tissue, and lowered the increase in heart weight due to hypertrophy	may be mainly due to the high protein content of sericin, flavonoids, and n-3 fatty acids that have potential free radical scavenging and antioxidant activities (the author speculated; needs to be experimentally confirmed)	[49]
hypolipidemic effect	1% NaCl solution extract of silk cocoons	New Zealand white rabbits /in vivo	cholesterol powder mixed with coconut oil	50 mg/ 100 g	reduced the levels of total cholesterol, triglycerides, and low-density lipoprotein, as well as the size of atherosclerotic plaque in the aorta; increased the high-density lipoprotein level and body weight	probably inhibited the second step of lipid implantation in the injured arterial wall by its lipid-lowering and antioxidant properties (the author speculated; needs to be experimentally confirmed)	[51]
	sericin	C57BL/6 mice /in vivo	fed with fat-rich diets	1000 mg/kg	increased lipid excretion in feces and restored intestinal wall morphometry in obese mice	NA	[11]
anti-inflammatory effect	silkworm cocoon-derived carbon dots	C57 black mice and Kunming mice /in vivo	(1) dimethylbenzene-induced ear oedema; (2) vascular permeability induced by acetic acid; (3) lipopolysaccharide- induced sepsis model	0.35, 0.7, and 1.4 mg/kg	significantly lowered the percentage inflammation at the doses of 0.7 and 1.4 mg/kg, and the plasma extravasation of the test groups was similar to that of the dexamethasone group	inhibition of the expressions of IL-6 and TNF-α	[3]
	sericin	Sprague–Dawley rats /in vivo	imiquimod-induced skin psoriasis	2.5, 5, and 10% sericin cream applied topically	10% sericin had the desired effect of improving skin psoriasis, similar to that of betamethasone and calcitriol treatments	reduction in cytokine production of Th17 cells by interfering with the JAK-STAT signaling pathway; modulation of immune response via upregulation of galectin-3 and downregulation of sphingosine-1-phosphate lyase1	[12]
antioxidant effect	ethanolic extract of the green cocoons	in vitro	DPPH and ABTS assay	in DPPH test: IC50 = 296.95 ± 13.24 μg/mL; in ABTS test: IC50 = 94.31 ± 9.13 μg/mL	showed excellent antioxidation	NA	[57]
	diazo cocoon extracts	in vitro	DPPH and ABTS assay	NA	exhibited high antioxidant activities	NA	[58]
	silk sericin and associated secondary metabolites (polyphenols and flavonoids)	in vitro	human dermal fibroblast cells	NA	the human dermal fibroblast cells treated with silk sericin exposed to UVA1 showed a significant increase in total collagen content	upregulates the expression of MMP-1 in human dermal fibroblast cells along with MMP-3, resulting in the degradation of collagen, and leads to the loss of the structural integrity of the skin	[59]
antiviral and antimicrobial effect	95% ethanol extract of silk cocoon	in vitro	HSV-1 and HSV-2	NA	the inactivation of HSV-1 and HSV-2	drastically reduced HSV-induced cell death and prevented inflammation by reducing the production of inflammatory cytokine genes	[21]
	silkworm cocoon	in vitro	three different species of fungi: *Candida albicans, Beauveria bassiana*, and *Saccharomyces cerevisiae*	NA	strongly suppressed the sporular growth of the three fungal species	BmSPI51 attaches to mannan and β-D-glucan on the surface of fungal cells, thus inhibiting fungal growth	[61]

NA: not available.

### 3.7. Other Effects

SC has other pharmacological activities that include antitumor and analgesic effects. Its primary component, SS, has an effect against colon cancer cell lines like SW480, where it regulates cellular processes, such as apoptosis, by increasing caspase-3 activity and reducing Bcl-2 expression, highlighting its therapeutic potential in tumor treatment [62]. Moreover, the unique strength and elasticity of SC make it well suited for treating tendon strains and aiding wound healing. It also possesses pain-relieving properties. Although this effect has been observed in tendinopathy models, further research is required to reveal its mechanisms of action [63].

## 4. Practical Applications from Laboratories to Clinics and Markets

As has been mentioned previously, SF, SS, and their related derivative materials are the primary bioactive components of SC, and they also act as biodegradable carriers with good compatibility. They have been widely used in drug delivery systems, tissue engineering, regenerative medicine, and in vitro diagnostics. These applications offer new treatment options for a variety of medical conditions. Nevertheless, most studies are still in the laboratory investigation stages, and only a few products have entered clinical trials and even the market. The current studies and the marketed products are primarily based on SF. The safety and efficacy of SF-based products have been confirmed through clinical studies, and a few of them have been approved by the Food and Drug Administration (FDA). The continuous development of SF bodes well for the broad prospects of silk-based materials in biomedical applications and markets.

### 4.1. Laboratory Investigations

The wide range of applications of SF and its derived materials has demonstrated significant innovation in the laboratory research stage, and these materials provide advanced solutions for future clinical applications and medical diagnostics.

#### 4.1.1. Application in Drug Delivery Systems

SF has been used in drug delivery systems to act as a promising carrier of films [64,65], nanoparticles [66,67], hydrogels [68], and microneedles [69,70,71,72] for diverse therapeutic attempts. The schematic diagram is depicted in Figure 1.

SF films loaded with honeysuckle flower extract induced apoptosis in HeLa cells, signifying its potential use as a promising material for cancer therapy [64]. SF films also act as a sustained release delivery system for insulin-like growth factor-1 that significantly accelerates wound healing in diabetic mice. The zero-order release kinetics of SF films ensure the continuous presence of insulin-like growth factor-1 at the wound site, thus enhancing reepithelialization and granulation tissue formation [65].

The integration of SF into nanoparticle-based drug delivery systems offers a promising approach to enhancing the efficacy and safety of various therapeutic agents. To overcome the poor bioavailability of curcumin, SF nanoparticles have been engineered to encapsulate curcumin, leading to improved cellular uptake and cytotoxicity against cancer cells while sparing healthy cells [66]. Baicalein, which is a flavonoid with anti-inflammatory properties, has been combined with SF to enhance its therapeutic effects. The resulting SF–baicalein complex demonstrated significant anti-inflammatory activity in a zebrafish model, indicating potential for developing new anti-inflammatory drugs with reduced side effects [67]. These systems have shown the particular promise of SF in cancer therapy and anti-inflammatory applications, with potential for further expansion into other areas of pharmacotherapy.

SF hydrogels, with their unique properties, show promise as a carrier for protein drugs, offering a sustainable and effective solution for drug delivery systems. SF hydrogels have demonstrated good drug-loading capacities, with a cumulative release of 80% bovine serum albumin within 12 h. Additionally, they simultaneously exhibit excellent stability and biocompatibility, with a storage modulus that increases significantly with concentration. The degradation behavior of SF hydrogels in various media further confirmed their potential for broad applications in biomedicine [68].

SF microneedles have garnered considerable interest due to their biocompatibility, mechanical strength, and tunable degradation properties. They are typically prepared using a process that involves the degumming of SC, the dissolution of silk fibers, and the subsequent solidification into a microneedle form. SF microneedles have been successfully applied for the transdermal delivery of insulin, demonstrating the potential to provide a painless and effective alternative for diabetic patients [69]. Additionally, their use in vaccine delivery has shown promising results, with the ability to elicit robust immune responses against various pathogens. Figure 2 shows the preparation process for vaccine-coated SF microneedles [70]. The ability of SF microneedles to protect labile biomolecules and control release rates makes them suitable for a wide range of therapeutic applications [71,72], and this represents a significant advancement in the field of transdermal drug delivery and positions them as a versatile platform for the delivery of diverse therapeutic agents.

#### 4.1.2. Application in Tissue Engineering

The exceptional multifunctionality of SF has established it as a significant material in the field of biomedical tissue engineering [73,74]. It is widely used to construct human scaffolds and develop bone hydrogels, making it a leading candidate material for tissue engineering (as summarized in Figure 3).

SF-based scaffolds are commonly recognized for their superior biocompatibility and tunable mechanical properties. With the use of special catalysts, the precise control of the cryogelation process and catalytic cross-linking reactions has promoted the development of new SF scaffolds that surpass the structural and physical properties of traditionally prepared scaffolds, demonstrating better flexibility, elasticity, and cellular biological performance [75]. Furthermore, the combination of SF with hyaluronic acid [76] and silver nanoparticles [77] has been shown to be particularly useful in bone tissue engineering. By adjusting the proportions and cross-linking strategies of these materials, the developed composite scaffolds have shown significant advantages in promoting cell proliferation and biocompatibility and are especially suitable for cartilage and bone tissue engineering [78,79]. Furthermore, the use of 3D-printing technology to construct complex three-dimensional structures required for tissue engineering with bio-inks made from these composite materials (as shown in Figure 4 [80]) can result in implants and scaffold structures suitable for more complex structural and functional applications [80,81]. It can also simulate the complex geometric shapes of natural tissues and organs [82], which is crucial for tissue engineering applications.

The development of SF bone hydrogels represents a significant advancement in tissue engineering. These hydrogels provide an ideal three-dimensional growth framework for cells by mimicking the natural environment of the extracellular matrix, and they feature an appropriate water content, controllable degradation rates, and good mechanical stability [83]. In particular, by improving the SF structure by combining physical and enzymatic crosslinking strategies, an SF matrix composite hydrogel with enhanced strength and elasticity was obtained. This development also improves the shortcomings of traditional SF hydrogels with uneven structures and poor mechanical properties [84].

#### 4.1.3. Application in Regenerative Medicine

The application of SF encompasses numerous aspects of regenerative medicine, including bone regeneration, vascular regeneration, and soft tissue regeneration, which benefit from the ability of SF to be designed into various forms, such as porous scaffolds and fibrous nets, to meet diverse medical needs [85] (as summarized in Figure 5).

In bone regeneration, an innovative study successfully developed a biological composite thin film by combining hydroxyapatite with SF [86] that enhanced bone conductivity and biocompatibility. It also showed significant potential in promoting bone tissue repair and regeneration, even with slightly reduced tensile strength, offering new material options.

The high design flexibility of SF also plays a crucial role in the domain of vascular regeneration and soft tissue regeneration. Researchers have constructed vascular grafts with excellent blood compatibility and good in-body tolerance using advanced purification, processing, and functionalization techniques. These SF vascular grafts not only mimic the mechanical properties of natural arteries, thus facilitating the rapid restoration of the endothelial cell layer, but also promote vascular remodeling by modulating local inflammatory responses [87]. They significantly reduce the tendency for thrombus formation and show a lower thromboprotein adsorption rate than traditional polytetrafluoroethylene materials, a key to maintaining the long-term patency of vascular grafts [88]. In addition, a new type of “bio-origami” material has been developed by integrating SS into biopolymer films with the aim to promote skin regeneration. These films that utilize different concentrations of SS exhibit outstanding physicochemical properties, mechanical strength, and antioxidant capacities [89].

#### 4.1.4. Application in In Vitro Diagnosis

SC and its derived materials have shown extensive potential in the field of in vitro diagnosis, and the latest research progress in biosensing platforms, cell cultures, signal molecule monitoring, blood type identification, and biocompatibility improvements has shown its value in improving diagnostic accuracy, convenience, and biomedical applications [90,91,92,93,94,95] (as summarized in Figure 6).

Gold nanoparticles (Au NPs) have important applications in biological imaging due to their unique physicochemical properties. Through the regulatory role of SF as a carrier, a biocompatible Au NP with uniform size and good dispersion was prepared. The SF fibers played multiple roles in the formation of the Au NPs, including as a reactive substrate, reducing agent, and modifier. These Au NPs not only exhibit excellent performance in biological imaging but also serve as efficient contrast agents to improve the accuracy of medical imaging [90].

A 3D cell-adhesive sensing matrix was designed and fabricated from SC-derived hierarchical carbon fiber networks assembled with the iron porphyrin of hemin. The matrix, which provided excellent biocompatibility for cell adhesion and long-term culture, was capable of highly sensitive and selective monitoring of cell-released nitric oxide molecules that could be used for live-cell assays and to explore physiological processes in complex biological systems. This is of great significance for understanding cellular behavior and disease mechanisms [91].

A versatile and thermally stable immunosensing platform was investigated using natural SC membranes as the substrate material. The platform utilized the intrinsic properties of the SC membrane for the directional immobilization of biomolecules through immunoaffinity recognition. This not only improved the sensitivity and selectivity of the sensor but also enhanced the stability and reliability of the sensor by maintaining the biological activity of antibodies. The immunosensor showed significant detection capabilities and excellent stability for immobilized antibodies, making it suitable for immunoassays and potential applications in resource-limited settings [92]. Correspondingly, the natural SC membranes were also used to develop a rapid and reliable immunoassay for ABO and RhD blood group typing. The assay was simple, requiring only a few pipetting steps and observable results within 30 s, making it suitable for point-of-care testing. The accuracy of the assay was validated against the gold-standard tube test, and it showed a 100% match, suggesting its effectiveness for blood typing [93].

Another study explored the biofunctionalization of biomimetic silk biomaterials with recombinantly expressed domain V of the human basement membrane proteoglycan perlecan for blood-compatible surfaces. The recombinantly expressed domain V was covalently immobilized on SF using plasma immersion ion implantation, a method that does not rely on specific amino acids in the silk protein chain. The biomimetic silk biomaterials demonstrated blood compatibility and could be a promising platform for developing blood-contacting devices. This provides new strategies for the development of vascular transplantation and blood contact devices [94].

A visually detectable H_2_O_2_ sensing system was created based on a gold nanozyme-SF hydrogel hybrid. The hydrogel exhibited a fast response and high sensitivity for H_2_O_2_ detection, with excellent stability and selectivity. The system has significant potential for the clinical diagnosis of H_2_O_2_, showcasing compatibility with biological tissues and minimal cytotoxicity [95].

The application of SC and its derived materials in in vitro diagnoses has demonstrated its multifunctionality and biocompatibility, providing new possibilities for developing new diagnostic tools and improving disease diagnosis efficiency.

### 4.2. Clinical Studies

Apart from the favorable laboratory research results mentioned above, several products based on SF have entered different stages of clinical research. By searching the website of www.clinicaltrials.gov launched by the U.S. National Institutes of Health [96], we summarized the information of these clinical trials that can be seen in Table 3. Among these clinical trials, three trials have already been completed, while the rest are still under recruitment or have an unknown status.

NanoSilk Cosmo is a novel SF-based product developed with the aim of addressing the biophysical parameters associated with skin aging. This product has been the subject of clinical trials and research studies to evaluate its safety and efficacy for human facial skin, particularly for improving skin resiliency and hydration. The results of the study suggested that NanoSilk Cosmo is safe for use on human facial skin. Furthermore, it has been observed to improve skin resiliency and hydration, which are essential factors in maintaining youthful-looking skin. This indicates it may offer benefits in the management of skin aging, providing a potential new option for individuals seeking to improve the appearance and health of their skin [97].

The efficacy and safety of wound dressings are critical factors in the management of split-thickness skin graft donor sites. A wound dressing containing SF with a bioactive coating layer was developed as an alternative to traditional medicated paraffin gauze dressings. Clinical studies have revealed that SF-based wound dressings can accelerate the healing process, reduce pain associated with wound care, and improve the overall cosmetic outcome compared to traditional dressings such as medicated paraffin gauze. It been shown that these dressings reduce the time to complete wound closure, decrease the frequency of dressing changes, and minimize the risk of complications such as infection and scarring. The use of a wound dressing containing SF with a bioactive coating layer represents a promising advancement in the treatment of split-thickness skin graft donor sites [96].

### 4.3. Products on the Market

In recent years, with the significant progress in the application of SF in the medical field, multiple SF-based products have been successfully launched domestically and internationally, and they are summarized in Table 4. The most representative products are Silk Voice^®^, SERI^®^, Tympasil^®^, and Sidaiyi^®^.

Developed by Sofregen Medical, Inc., Silk Voice^®^ is the first and only natural silk protein injectable for tissue bulking. The FDA cleared it for vocal fold medialization and vocal fold insufficiency that may be improved by injection of a soft tissue bulking agent [98]. It comprises porous bioabsorbable silk particles suspended in an isotonic, aqueous formulation of cross-linked, high-molecular-weight hyaluronic acid. This unique composition enables the product to be biocompatible and easily integrated into the body’s tissues, with the silk particles gradually being absorbed over time as they promote the patient’s own tissue growth [99]. The development and approval of Silk Voice^®^ represent a milestone in the application of silk-based biomaterials in medical treatments. Another FDA-approved SF-based product, SERI^®^, was also developed by Sofregen Inc. It is a medical device used for abdominal wall reconstruction and plastic surgery, and it has shown high satisfaction and minimal complications in clinical trials [100].

Other countries have also approved the use of SF in vivo by their regulating authorities. Tympasil^®^ is a medical device produced by Daewoong-Bio, South Korea. This product is a thin, transparent patch made from SF that is designed for the repair of acute tympanic membrane perforations [101], offering a noninvasive and effective treatment option for patients. In clinical studies, Tympasil^®^ has demonstrated promising results in treating patients with chronic tympanic membrane perforations. The patch is softened by immersion in warm saline and then carefully placed over the perforation site. This process allows the patch to conform to the shape of the eardrum, promoting healing and regeneration of the tissue [102]. In China, SF has been approved for use as a clinical wound dressing, with the brand name of Sidaiyi^®^. It is designed as a two-layered system, featuring an SF sponge attached to a silicone membrane. This unique construction is intended to provide an environment conducive to wound healing, offering protection from external contaminants while simultaneously promoting the regeneration of skin tissue [103]. It represents a significant advancement in the field of regenerative medicine and wound care. Moreover, other SF-related products have been approved in succession. For example, absorbable SF repair film is used for oral dental implant repair, and it can promote the growth of alveolar bone and prevent soft tissue invasion. It has become the first domestically approved Class III implantable SF medical device product.

## 5. Safety

The collective findings from the current studies underscore the potential of SF and SS as safe and effective materials for various applications. The safety assessment of SS was studied using a series of rigorous tests that included bacterial reverse mutation tests, mammalian erythrocyte micronucleus tests, and a 90-day subchronic toxicity study in Sprague–Dawley rats. SS exhibited no genotoxicity or mutagenicity, and its no-observed-adverse-effect level was determined to be 1 g/kg/day, suggesting a low toxicity profile [104].

Another study examined the toxicological assessment and potential allergy risks associated with SF. A comprehensive battery of tests, such as the Ames test, the in vivo mouse erythrocyte micronucleus test, and 28-day oral toxicity studies, were conducted to evaluate the safety of SF for consumption. An in vitro pepsin digestion assay was performed to assess the allergenicity of SF [105]. The studies concluded that SF does not raise concerns regarding mutagenicity, genotoxicity, toxicity, or allergenicity, making it a promising material for food preservation applications.

The cytotoxicity, biocompatibility, and cell adhesion kinetics of SF were further explored. It was found to be immunologically inert, invoking minimal immune responses. This was evaluated by measuring the nitric oxide and factor-alpha production in murine peritoneal macrophages and RAW 264.7 murine macrophages, and it showed a comparable stimulation as collagen. Cell viability studies and flow cytometric analyses indicated that the SF matrices supported cell growth and proliferation comparable to collagen, making it suitable for long-term culture applications [106].

## 6. Conclusions and Future Perspectives

Researchers in the field of medical bioengineering are especially interested in SC due to its multifunctionality, morphological flexibility, and excellent biocompatibility. The dual functionality of SC both as a traditional Chinese medicine and the raw material of biocompatible carriers presents an intriguing avenue for its further exploration and utilization.

Nevertheless, the research and application of SC is still facing a raft of issues and challenges: (1) Existing research on the pharmacological effects and clinical applications of SC remains in the initial phases, necessitating more in-depth studies on its mechanisms and therapeutic efficacy. (2) Traditional silk biomaterial preparation involves dissolution and reshaping, a complex and lengthy process that calls for the development of simpler and more sustainable methods. (3) SC, as a potential wound dressing material, inherently possesses certain antimicrobial properties. However, it may need to be improved or combined with other materials to enhance the antimicrobial effectiveness to meet higher medical application standards. (4) SC is a natural material with unique geometric characteristics and can be used as an excellent model for biomimetic design and applications. Nevertheless, effectively translating these characteristics into practical applications remains challenging. (5) More comprehensive research is required on the mechanical behavior of SC-derived materials, especially their performance under different strain rates and microstructures, to optimize their use in various engineering applications. (6) Wild SC is a potential alternative for SC resources, but effective utilization of these resources and overcoming challenges related to silk extraction and quality compared to domesticated silkworms remain issues that need to be addressed.

To summarize, as our understanding of SC and its primary components deepens, we believe that its latent value and application scope will continue to be explored and expanded. This will bring more surprises and breakthroughs to future medical and life science research. Notably, the research and application of SC encompass a wide range of issues that require interdisciplinary collaboration, while existing challenges should be overcome through innovative approaches.

## Figures and Tables

**Figure 1 pharmaceuticals-17-00817-f001:**
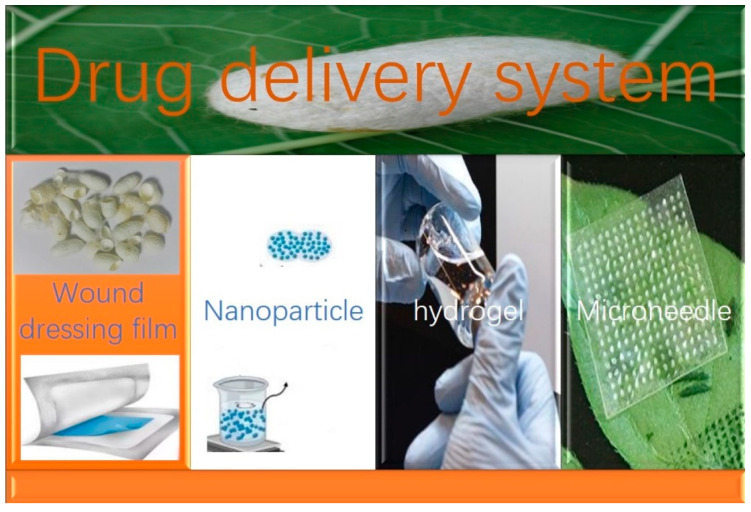
The application of SC-derived materials in drug delivery systems.

**Figure 2 pharmaceuticals-17-00817-f002:**
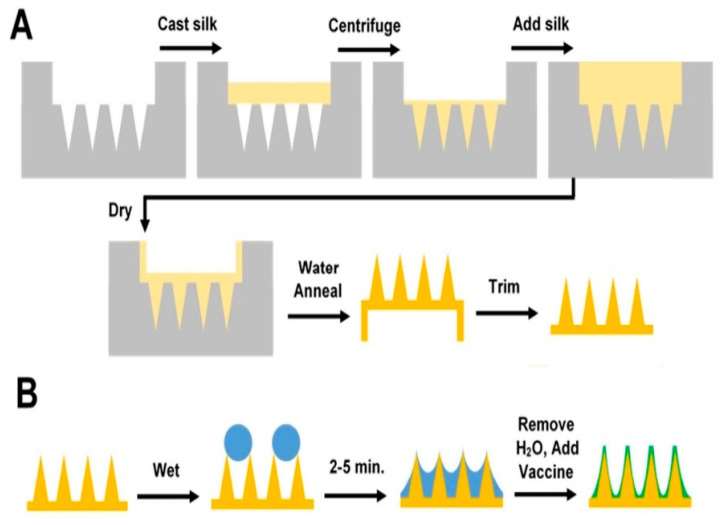
The preparation process for vaccine-coated silk fibroin microneedles. (**A**) Silk microneedles were demolded, annealed with water, and trimmed to create the final uncoated constructs. (**B**) For uniform coating of silk microneedles with vaccine-containing solution, the devices were pre-wetted with water droplets.

**Figure 3 pharmaceuticals-17-00817-f003:**
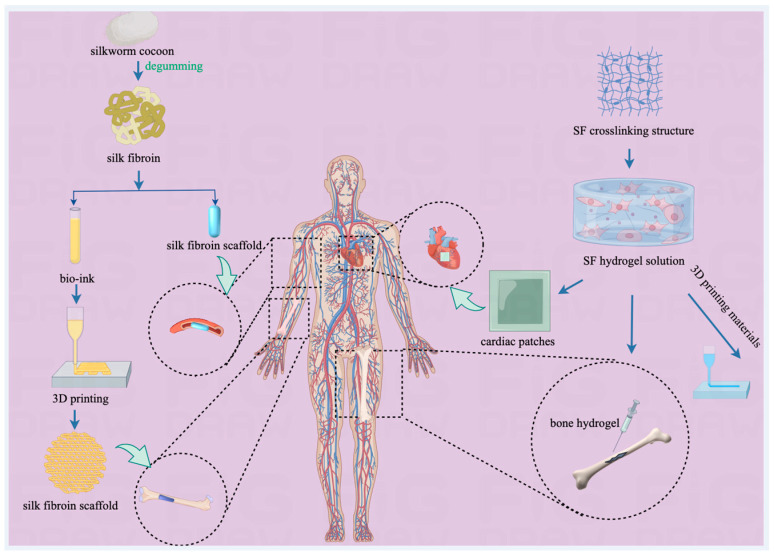
The application of SC-derived materials in tissue engineering.

**Figure 4 pharmaceuticals-17-00817-f004:**
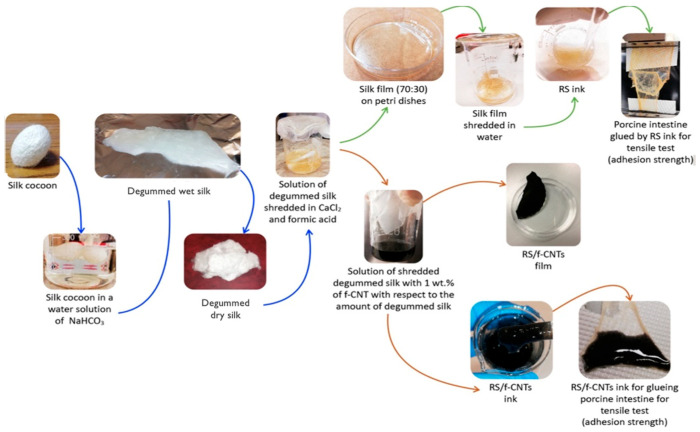
Schematic diagram of a solvent-free method for preparing a bio-ink of SF extracted from SC.

**Figure 5 pharmaceuticals-17-00817-f005:**
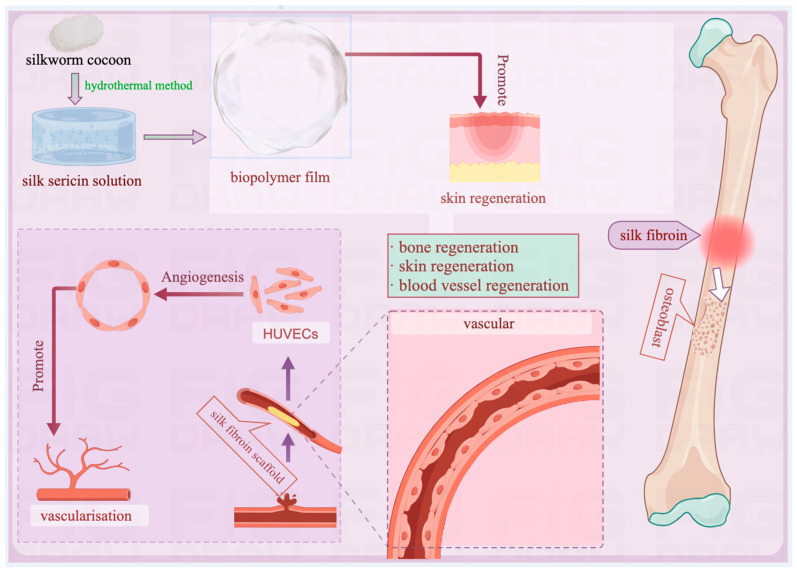
The application of SC-derived materials in regenerative medicine.

**Figure 6 pharmaceuticals-17-00817-f006:**
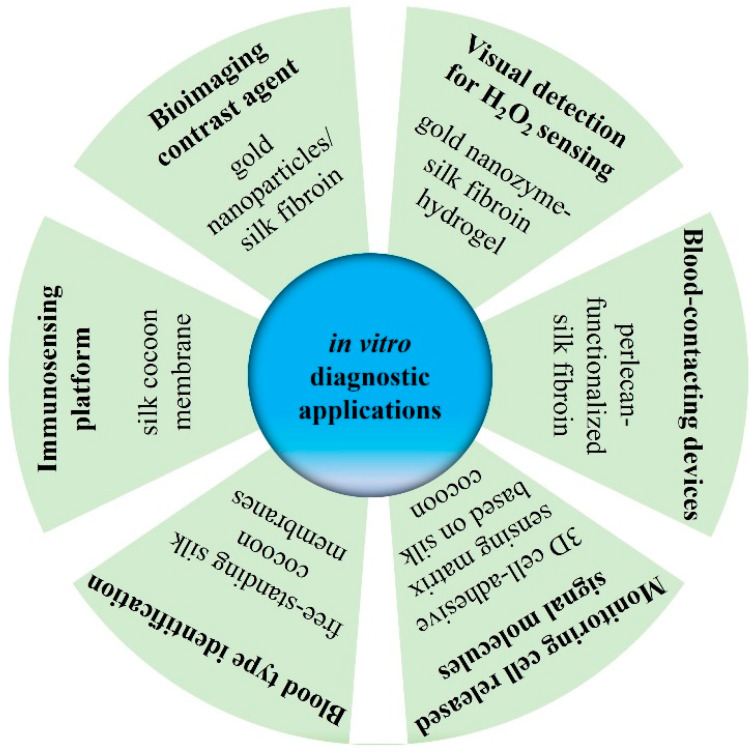
The application of SC-derived materials in in vitro diagnosis.

**Table 1 pharmaceuticals-17-00817-t001:** Main bioactive compounds extracted from SC.

Main Compound	Extraction Method	Content
silk fibroin	soda boiling process (degumming)	70–80%
silk sericin	hydrothermal method; chemical method; enzymatic method	20–30%
flavonoids	solvent extraction process	<1%
calcium oxalate	acid dissolution method	
other peptides	enzymolysis method	

**Table 3 pharmaceuticals-17-00817-t003:** Clinical trials of silk fibroin products [96].

Official Title	Sponsor	Enrollment	Objective	Dosage	Study Phase	Condition	Location	Current Status
A Comparative Evaluation of Subgingivally Delivered Chlorhexidine, Silk Fibroin and Combination of Fibroin and Chlorhexidine as Local Drug Delivery in Periodontitis—A Randomized Control Trial	Krishnadevaraya College of Dental Sciences & Hospital	15 (Estimated)	To evaluate the effect of silk fibroin as a drug delivery system while simultaneously assessing the efficacy of silk fibroin in comparison to chlorhexidine	Films	Phase 1	Periodontal Pocket	Not provided	Not Yet Recruiting
Efficacy and Safety of Wound Dressing Containing Silk Fibroin With Bioactive Coating Layer Versus Medicated Paraffin Gauze Dressing in the Treatment of Split-thickness Skin Graft Donor Sites	Chulalongkorn University	29 (Actual)	To compare wound dressing containing silk fibroin with bioactive coating layer with Bactigras^®^, with regard to healing time, patients’ pain intensity, skin’s transepidermal water loss after healing and evidence of infection in the treatment of split-thickness skin graft donor sites	Films	Phase 1 Phase 2	Impaired Wound Healing; Infection of Skin Donor Site; Late Complication From Skin Graft; Intractable Pain	Thailand	Completed
Manufacturing, Characterization and Evaluation of the Effect of Silk Fibroin Membranes, Loaded or Not With Neurotensins on Open Wounds in the Palate: Randomized Clinical Study	Universidade Estadual Paulista Júlio de Mesquita Filho	66 (Estimated)	To manufacture and characterize silk fibroin membranes loaded or not with neurotensin and to evaluate clinical, patient-centered, and immunological parameters to determine the effect of using these membranes on open wounds on the human palate	Films	Not Applicable	Wound Healing; Palate Wound	Brazil	Recruiting
A New Drug Delivery System—Silk Fibroin Film Loaded or Not With Insulin on Palatal Mucosa Wound Healing: in Vitro Study and a Randomized Clinical Trial	Universidade Estadual Paulista Júlio de Mesquita Filho	75 (Estimated)	To evaluate the effect of silk fibroin films loaded or not with insulin in the repair of palatal mucosa open wounds	Films	Not Applicable	Wound Healing; Palate Wound	Brazil	Unknown
A Pilot Study to Evaluate the Reconstruction of Digital Nerve Defects in Humans Using an Implanted Silk Nerve Guide	Silk Biomaterials srl	4 (Actual)	To ascertain the feasibility and safety of the procedure using SilkBridge—a biocompatible silk fibroin-based scaffold—for the regeneration of sensory nerve fibers	Scaffold	Not Applicable	Peripheral Nerve Injury Digital Nerve Hand	Switzerland	Unknown
NanoSilk Cosmo: Evaluation of a Novel Silk Complex on Biophysical Parameters Related to Skin Aging	University of Colorado, Denver	46 (Actual)	To evaluate a novel silk complex on biophysical parameters related to skin aging including skin resilience, elasticity, and hydration	Nanosolution	Not Applicable	Aging	United States	Completed
Multi-center, Randomized, Active-controlled, Single-blind, Parallel Two-group Trial of HQ^®^ Matrix Soft Tissue Mesh and ULTRAPRO^®^ Partially Absorbable Lightweight Mesh for the Treatment of Inguinal Hernia	Zhejiang Xingyue Biotechnology Co., Ltd.	144 (Estimated)	To evaluate the safety and effectiveness of HQ^®^ Matrix Soft Tissue Mesh for the Treatment of Inguinal Hernia	Scaffold	Not Applicable	Inguinal Hernia	China	Unknown
Pilot Evaluation of Cosmetic Outcome and Surgical Site Infection Rates of Coated VICRYL* Plus Antibacterial (Polyglactin 910) Suture Compared to Chinese Silk in Scheduled Breast Cancer Surgery	Ethicon, Inc.	101 (Actual)	To evaluate the cosmetic outcome and surgical site infection in approximately 100 patients from 6 centers in China undergoing scheduled modified radical mastectomy for breast cancer	Silk Suture	Phase 4	Breast Cancer	China	Completed

**Table 4 pharmaceuticals-17-00817-t004:** Silk fibroin products on the market.

Product Name	Main Compositions	Indications	Approval Year	Nation
Silk Voice^®^	A silk fibroin injection	To treat vocal cord-mediated and vocal cord dysfunction.	2018	USA
SERI^®^	Surgical stent device based on silk fibroin	Abdominal wall reconstruction and plastic surgery.	Unknown	USA
Tympasil^®^	A silk fibroin patch	To treat ear drum perforations.	Unknown	Korea
Antibacterial wound dressing patch	A silk fibroin patch	Postoperative incisions, skin surface abrasions, and ulcer coverage.	2020	China
Functional healing wound dressing patch	Silk fibroin, surface coated with a composite silicon-based powder	Promoting healing, repair, and coverage of postoperative wounds, abrasions, and non-healing wounds.	2021	China
Absorbable silk fibroin repair film	Composed of silk fibroin, glycerol, and water	Used in conjunction with bone meal as a physical barrier for preserving the extraction site of adult patients after tooth extraction.	2022	China
Silk fibroin film dressing	Silk fibroin with an amino acid content of ≥90%	For skin area coverage.	2020	China
Silk fibroin vaginal packing gel	Gel consists of silk fibroin, carbomer, triethanolamine, sodium ethyl paraben, glycerin, and purified water	To block HPV infection in the reproductive tract and prevent cervical lesions caused by HPV infection. To improve the vaginal microenvironment, alleviate itching, pain, congestion, edema, increased secretion, purulent discharge symptoms caused by chronic cervicitis, and reduce the surface of cervical erosion.	2023	China
Silk fibroin hydrogel dressing	Gel composed of silk fibroin and purified water	Coverage and care of non-chronic wounds after laser surgery.	2022	China
Light guide gel	Composed of carbomer 940, glycerol, silk fibroin, pentanediol, sodium hydroxide, and purified water	For thermal insulation and light guidance during photon therapy, in conjunction with photon therapy equipment.	2022	China
Liquid dressing	A solution composed of silk fibroin, sodium alginate, sodium carboxymethyl cellulose, glycerol, and purified water	For the care of superficial wounds and surrounding skin such as small wounds, abrasions, and cuts.	2021	China
Liquid wound dressing	A solution composed of silk fibroin, sodium chloride, glycerol, and carbomer	For the care of superficial wounds and surrounding skin such as small wounds, abrasions, and cuts.	2020	China
Liquid dressing	A solution composed of silk fibroin, glycerol, sodium benzoate, potassium sorbate, and carboxymethyl cellulose	For the care of superficial wounds and surrounding skin such as small wounds, abrasions, and cuts.	2020	China

## Data Availability

No data were used for the research described in the paper.

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
