# Peer review of "Silkworm Cocoon: Dual Functions as a Traditional Chinese Medicine and the Raw Material of Promising Biocompatible Carriers"

_pharmaceuticals, 2024, doi:10.3390/ph17070817_

Round 1
Reviewer 1 Report
Comments and Suggestions for Authors
The manuscript ID: pharmaceuticals-3061914 entitled “Silkworm Cocoon: Dual Functions as a Traditional Chinese Medicine and the Raw Material of Promising Biocompatible Carriers” by Tian et al. can be accepted to publication in Pharmaceuticals after minor revision.
Specific comments
In the present review paper the Authors described the multiple functions and applications of the silkworm cocoon (SC) in traditional Chinese medicine as well as in medical and biomedical fields where it is used as a raw material for biocompatible carriers. Thus, the analysis of its chemical compositions and various biological activities, including hypoglycemic, cardioprotective, hypolipidemic, anti-inflammatory, antioxidant, and antimicrobial actions, as well as its applications in drug delivery systems, tissue engineering, regenerative medicine, and in vitro diagnostics were deeply discussed. The authors highlighted that the biodegradability and excellent biocompatibility of SC to develop new medical materials and treatment methods which increases significantly its possibility to use in the contemporary medical domain. The aim of this review was not only to show the current research status of SC but also to indicate the further development and application of SC-based products.
In my opinion, the topic of the paper can be interesting for the readers, the manuscript is well written and is based on current literature. This review paper can be published in Pharmaceuticals after minor revision.
1. The Authors presented in graphical form the applications of SC in drug delivery systems, tissue engineering, regenerative medicine, and in vitro diagnostics. On the other hand, there are no original figures from the References decribed in the manuscript. In my opinion, the Authors should selected the most representative for described issues and they (2 or 3) should be also included to the paper.
2. Editorial mistake:
a) Page 8 line 222: 96.95±13.24 μg/mL and 94.31±9.13 μg/mL
b) Page 18; line 526: 2)Tr
Author Response
Reviewer 1:
- The Authors presented in graphical form the applications of SC in drug delivery systems, tissue engineering, regenerative medicine, and in vitro diagnostics. On the other hand, there are no original figures from the References decribed in the manuscript. In my opinion, the Authors should selected the most representative for described issues and they (2 or 3) should be also included to the paper.
Answer: Thank you for your positive feedback on our article. According to your suggestion, we have selected two representative figures (please see Figure 2 and Figure 4) from the references to make the content of the article more substantial.
- Editorial mistake:
a) Page 8 line 222: 96.95±13.24 μg/mL and 94.31±9.13 μg/mL
Answer: We appreciate your valuable feedback. In response to your suggestion, we have carefully changed 296.95±13.24 μg/mL and 94.31±9.13 μg/mL to 296.95 ± 13.24 μg/mL and 94.31 ± 9.13 μg/mL in line 228. The appropriate space has been added at the right positions.
b) Page 18; line 526: 2)Tr
Answer: Thank you for your kind suggestion. We have changed “2)Tr” to “2) Tr” as seen in Line 552. The appropriate space has been added at the right positions.
Reviewer 2 Report
Comments and Suggestions for Authors
many points must be improved
1- please add methods of extraction sc, ss, sf in the diagram
2- add a table containing the most bioactive compounds extracted from Sc
3- line 102, In addition to primary proteins like SF and SS, SC is composed of peptides ( not understood)
4- the head order of Biological activities-1.1, 1.1. and so on please review
Author Response
many points must be improved
1. please add methods of extraction sc, ss, sf in the diagram.
Answer: Thank you for your advice. We have added the extraction methods of SF and SS, as shown in Figure.3 and Figure.5, respectively, which can be seen in the revised manuscript.
2. add a table containing the most bioactive compounds extracted from Sc.
Answer: Thank you for your suggestion. We have added Table 1, which contains the most bioactive compounds extracted from SC.
3. line 102, In addition to primary proteins like SF and SS, SC is composed of peptides ( not understood).
Answer: We are sorry for the misunderstanding. In addition to primary proteins like SF and SS, SC is composed of peptides with different sequences and functions. These peptides are small molecular protein fragments composed of amino acids and are mainly found in the degradation products of SF or SS. We have added this information in the revised manuscript, which can be seen in Line 107-109.
4. the head order of Biological activities-1.1, 1.1. and so on please review.
Answer: Thank you for your advice. We have reviewed the heading order throughout the manuscript and confirmed that the changes are correct.